# Physical Activity and Quality of Life among High School Teachers: A Closer Look

Danijela Živković [1], Ljubica Milanović [2], Anđela Đošić [1], Ana-Maria Vulpe [3,*], Tijana Purenović-Ivanović [1], Milan Zelenović [2], Dragoș Ioan Tohănean [4,*], Saša Pantelić [1], Constantin Sufaru [3] and Cristina Ioana Alexe [3]

[1] Faculty of Sport and Physical Education, University of Niš, 18000 Niš, Serbia; danijela.zivkovic@fsfv.ni.ac.rs (D.Ž.); andjela.djosic@fsfv.ni.ac.rs (A.Đ.); tijana@fsfv.ni.ac.rs (T.P.-I.); panta@fsfv.ni.ac.rs (S.P.)

[2] Faculty of Physical Education and Sports, University of East Sarajevo, 71123 East Sarajevo, Bosnia and Herzegovina; ljubica.milanovic@ffvis.ues.rs.ba (L.M.); milan.zelenovic@ffvis.ues.rs.ba (M.Z.)

[3] Department of Physical Education and Sports Performance, "Vasile Alecsandri", University of Bacău, 600115 Bacău, Romania; sufaruconstantin@ub.ro (C.S.); alexe.cristina@ub.ro (C.I.A.)

[4] Faculty of Physical Education and Mountain Sports, "Transilvania" University of Brașov, 600115 Brașov, Romania

[*] Correspondence: zaharia.ana@ub.ro (A.-M.V.); dragos.tohanean@unitbv.ro (D.I.T.)

**Abstract:** Background: Understanding the relationship between teachers' physical activity (PA) and quality of life (QoL), which is impacted by work-related stress, could help develop guidelines for improvement. The purpose of this study was to investigate the impact of physical activity on high school teachers' quality of life and the differences in QoL and PA between male and female teachers. Methods: The sample consisted of 499 respondents (193 men and 306 women), all working in the educational system. The International Physical Activity Questionnaire (short form) was used for PA assessment, and the WHOQoL questionnaire to measure QoL. Results: Physical health and Psychological health domains were areas where male teachers scored better ($p < 0.01$, both), while female teachers had higher scores in Social relationships domain ($p < 0.05$). Regression analysis showed that PA affects Physical health: Sig. = 0.056; Psychological health: Sig. = 0.000; Social relationships: Sig. = 0.001; Environment: Sig. = 0.021 in men, and Physical health (Sig. = 0.009) and Psychological health (Sig. = 0.039) in women. Conclusions: The findings of this study allow us to conclude that, whereas female teachers' PA primarily impacts their physical and psychological domain, male teachers' PA has an impact on their overall QoL.

**Keywords:** levels of physical activity; teachers; physical health; psychological health; social relationships; environmental domain of quality of life





## 1. Introduction

The teaching profession is becoming increasingly challenging as a consequence of the growing number of work assignments and the constant need for additional education (Hargreaves et al. 2007). This profession is one of the occupations with the highest level of stress at work (Travers 2017), which significantly affects teachers' mental health (Guerrero-Barona et al. 2018) and quality of life (Hong et al. 2003). On the other hand, most teachers lead a sedentary lifestyle, which stems from the specifics of the work process (Rosales-Ricardo et al. 2017). A sedentary lifestyle, as well as physical inactivity, represent a major public health problem (Nooijen et al. 2019). Based on the most current recommendations, individuals should engage in at least 150 min a week of moderate-intensity or 75 min a week of vigorous-intensity physical activity (WHO 2020). Engaging in the aforesaid quantity of physical activity can help avoid a number of health conditions, such as heart disease, type 2 diabetes, obesity, and different malignancies (Janssen and LeBlanc 2010). Also, regular physical activity significantly reduces the risk of developing anxiety and

depression and improves body image (Mammen and Faulkner 2013; Krzepota et al. 2015; Milanović et al. 2022) in the healthy adult population (Bernard et al. 2018; White et al. 2020). Conversely, a lifestyle that is primarily sedentary is linked to a higher risk of all the diseases listed above as well as premature mortality (Buckley et al. 2015).

Although there is knowledge about the benefits of PA and the health consequences of physical inactivity, the adult population does not meet the guidelines given by the World Health Organization (WHO 2020). More specifically, a 2018 global survey revealed that over 25% of adults are insufficiently physically active. These data indicate that more than 1.4 billion adults are at risk of developing various diseases caused by physical inactivity (Guthold et al. 2018). The problem is even greater considering that adults spend a large part of their lives at work, with the majority of the population having jobs that involve a sedentary lifestyle (Buckley et al. 2015). According to epidemiological studies, office workers spend two-thirds of their working lives seated (Ryan et al. 2011; Parry and Straker 2013). Moreover, the majority of contemporary businesses require computer expertise, and it has swiftly become the norm for employment worldwide (Badr et al. 2021). Teaching involves a lot of sitting due to reviewing homework and preparing lessons on the computer, administrative work, as well as other activities that do not require physical activity. An additional element that leads to the emergence of health problems in teachers is their prolonged standing during teaching (Durmus and Ilhanli 2012).

Given that regular PA is a significant factor in determining one's quality of life, spending too much time in a sedentary position lowers the quality of life of working people (Bernard et al. 2018). The World Health Organization defines quality of life as "individuals' perceptions of their position in life in the context of the culture and value systems in which they live and in relation to their goals, expectations, standards, and concerns" (Whoqol Group 1995). It is a broad concept that intricately encompasses a person's state of mind, physical health, degree of independence, social connections, personal values, etc. Marquez et al. (2020) found that physical activity significantly affects the quality of life of individuals between the ages of 18 and 65. Research findings showed that while various physical activity programs increase the working population's quality of life in every domain, the most important factor affecting quality of life is the frequency of physical activity. According to a study by Brodáni and Žišková's (2015), which included a sample of female teachers, the most physically active female teachers, those in the 36–46 age range, have the greatest correlations between their weekly physical activity and quality of life. Compared to those who are less physically active, working people who engage in regular physical activity have a higher quality of life (Krzepota et al. 2015; Puciato et al. 2017, 2018; Đošić et al. 2021; Patten et al. 2023). Previous studies have established the relationship between physical activity and adults' quality of life, with men reporting a higher level of quality of life than women (Sławińska et al. 2013). In addition to the aforementioned factors, research has shown a correlation between moderate to high levels of physical activity and particularly mental health in males; however, no such link was found in females (Saridi et al. 2019).

According to Rongen et al. (2014), employees who follow World Health Organization guidelines and engage in physical activity not only benefit from improved health and a higher quality of life, but also perform better at work than their inactive counterparts. Studies have also found that children whose teachers were physically active had a higher level of physical activity than children whose teachers did not exercise physical activity, both in early childhood (Cheung 2020) and in adolescence (McDavid et al. 2012). According to the recent systematic review (Marquez et al. 2020), most of the research shows a positive impact of PA on quality of life. Nonetheless, not enough research has been performed regarding the effect of physical activity on the lives of men and women who work in the educational system. Therefore, the aim of the study was to determine whether total physical activity and its individual levels affect the quality of life of teachers, both male and female, as well as whether there are differences in the PA levels and domains of QoL in this population group.

## 2. Materials and Methods

### 2.1. Study Design and Procedures

Male and female high school teachers were asked to complete a questionnaire on their quality of life and physical activity levels as part of the study. Prior to completing the questionnaires, all participants received supplementary explanations and instructions from the researchers regarding the questionnaires and how to complete them. The participants were informed of the study's aims. The responders manually completed the questionnaire. There was no time restriction on answering the questions on the questionnaire. Each respondent had the option to discontinue participation in the study at any point while completing the survey; participation was entirely voluntary. The respondents received notice that their answers would remain anonymous and that the data would only be used for study. Questionnaires that were not completed were excluded from additional analysis. Sociodemographic questions were also included in the questionnaire. Body mass and height measurements were recorded prior to the respondents' completion of the questionnaires. The study was carried out in compliance with guidelines for research involving human participants and the Declaration of Helsinki (Christie 2000; World Medical Association 2008).

### 2.2. A Sample of Respondents

The sample of subjects consisted of a total of 499 respondents, of which 193 respondents were male teachers (37.4 ± 8.4 years old) and 306 female respondents (40.9 ± 8.6 years old). Average recorded BMI values were 27.1 ± 2.8 for men and 24.1 ± 3.2 for women (Table 1).

**Table 1.** Characteristics of the sample.

|  | Men (*n* = 193) | Women (*n* = 306) |
|---|---|---|
| Age [years] | 37.4 ± 8.4 | 40.9 ± 8.6 |
| Body Hight [cm] | 183.3 ± 5.9 | 168.2 ± 5.5 |
| Body Weight [kg] | 91.3 ± 13.1 | 68.1 ± 9.4 |
| BMI [kg/m$^2$] | 27.1 ± 2.8 | 24.1 ± 3.2 |

Individuals who fulfilled the following criteria were qualified to take part in the research: they had to be employed in the education sector, provide their consent to participate in the research before it began, be physically independent individuals aged between 24 and 65, and not be suffering from any serious health issues.

The criteria for exclusion from the study were: being in the recovery phase from some form of acute illness, being in the process of rehabilitation from injuries, and an age younger than 24 and older than 65 years.

### 2.3. Sample Size

To calculate the sample size, the analysis program G*power 3.1 (Faul et al. 2007) was used. The effect size was assumed to be (f2) = 0.15, alpha level 0.05, and power 80% (0.80), so the estimated total sample size was a minimum of 85 respondents per group.

### 2.4. A Sample of Measuring Instruments

All measurements that determined the characteristics of the sample (body mass, body height and Body Mass Index (BMI)) were carried out before the start of the survey in accordance with the recommendations by Eston and Reilly (2001). Height was measured using a Martin anthropometer (GPM, Switzerland) to the closest 0.1 cm. The measurement of body mass was performed using the digital scale Omron BF511; body mass (kg) divided by body height (m$^2$) produces a BMI (Weir and Jan 2019).

### 2.5. Physical Activity

The level of physical activity was determined using a self-assessment technique, the International Physical Activity Questionnaire in Short Form (IPAQ-Short Form) (Craig

et al. 2003). The short version of the IPAQ questionnaire contains seven questions and assesses physical activity in four domains: (1) high-intensity PA, (2) moderate-intensity PA, (3) low-intensity PA (walking), and (4) total PA and evaluates the level of physical activity in the last 7 days. Scores for every reported level of PA were computed using the metabolic equivalent (MET) in relation to minutes per day. By adding MET-minutes for each PA intensity level (walking/low intensity = 3.3 METs; moderate intensity = 4.0 METs; high intensity = 8.0 METs), the total weekly MET-minutes (MET min/week) were determined. For calculating the MET value, the pattern from the study was applied (Craig et al. 2003). Using the Ainsworth et al. (2011) Compendium, an average MET score was calculated for each type of activity. For instance, various forms of walking were considered together to establish an average MET value for walking. This same process was applied to determine average MET values for moderate-intensity activities and vigorous-intensity activities. The reliability and validity of the IPAQ-SF questionnaire has been established in previous studies (Battaglia et al. 2016; Tran et al. 2020).

*2.6. Quality of Life*

Quality of life was assessed using a shorter version of the World Health Organization's questionnaire—WHOQOL-BREF, which was internationally validated (Skevington et al. 2004). There are 26 items on the survey, which are broken down into 4 categories: Physical health, Psychological health, Social relations and Environment, and provides the possibility to determine individual scores for each of the categories. All questions within the domain were presented on a five-point Likert scale, where the options were (1) minimal, (2) little, (3) moderate, (4) very much, and (5) extreme. Higher values indicate a better quality of life. Numerous studies have proven the validity and reliability of the WHOQOL-BREF questionnaire (Hanestad et al. 2005; Ohaeri and Awadalla 2009; Ilić et al. 2019; Kalfoss et al. 2021)

*2.7. Statistical Data*

The mean and standard deviation (Mean $\pm$ SD), descriptive statistics basic elements were computed for each variable. A *t*-test was used to calculate differences in physical activity and quality of life domains between men and women. The Pearson correlation coefficient was used to find associations among physical activity and quality of life. Regression analysis was used to determine the impact of physical activity on teachers' quality of life. SPSS 20.0 was used as the statistical software to process all of the data (SPSS Inc., Chicago, IL, USA). The level of significance was set at 0.05.

**3. Results**

In regard to the findings (Table 2), it was concluded that female teachers have a significantly higher level of physical activity in high-intensity PA (Sig. = 0.014) compared to male teachers, while male teachers have a significantly higher level of physical activity in low-intensity PA (walking) (Sig. = 0.000). Additionally, the findings indicated that, while there was not a significant difference, male teachers were more physically active than female teachers in total PA. Quality of life was significantly higher in male teachers in the domains of Physical health (Sig. = 0.000) and Psychological health (Sig. = 0.000), while higher values were found in women in the domains of Social relationships (Sig. = 0.013) and Environment (NS between men vs. women).

**Table 2.** Differences between male and female teachers.

| | Men | Women | Men vs. Women | | 95% CI of the Difference | |
| --- | --- | --- | --- | --- | --- | --- |
| | | | t | Sig. | Lower | Upper |
| High intensity PA | 1002.73 ± 899.57 | 1185.60 ± 745.43 | 6.06 | 0.014 * | 1043.62 | 1186.39 |
| Moderate intensity PA | 1023.72 ± 794.15 | 1076.03 ± 932.26 | 0.41 | NS | 978.42 | 1133.25 |
| Low intensity (walking) PA | 1394.770 ± 660.52 | 1069.20 ± 795.49 | 22.54 | 0.000 ** | 1127.89 | 1261.85 |
| Total PA | 3412.32 ± 1089.40 | 3318.87 ± 1397.28 | 0.62 | NS | 3241.88 | 3468.01 |
| Physical health | 31.31 ± 2.31 | 30.11 ± 2.73 | 25.47 | 0.000 ** | 30.34 | 30.80 |
| Psychological health | 25.94 ± 2.09 | 24.57 ± 2.51 | 40.39 | 0.000 ** | 24.88 | 25.31 |
| Social relationships | 8.27 ± 1.08 | 8.50 ± 0.90 | 6.23 | 0.013 * | 8.33 | 8.50 |
| Environment | 30.81 ± 3.51 | 31.27 ± 3.73 | 1.83 | NS | 30.77 | 31.41 |

PA, Physical Activity in METs; NS, not significant; Sig., level of significance, ** $p < 0.01$, * $p < 0.05$; Mean—Mean value; SD—Standard Deviation.

Table 3 shows cross-correlations (Pearson's coefficient) between predictor and criterion variables for the sample of male and female teachers. The results of the cross-correlation of the level of physical activity and the quality of life of male teachers (Table 3) indicate that there are significant relationships between High-intensity PA and Psychological health (0.26; Sig. = 0.000) and High-intensity PA and Social relationships (−0.20; Sig. = 0.000), between Moderate-intensity PA and Social relationships (0.17; Sig. = 0.022), Moderate-intensity PA and Environment (−0.21; Sig. = 0.003), as well as Total PA and Psychological health (0.21; Sig. = 0.003) and Total PA with Environment (−0.18; Sig. = 0.014).

**Table 3.** Relationships of Physical Activity Levels and Quality of Life.

| | | PhysicalHealth | | Psychological Health | | Social Relationships | | Environment | |
| --- | --- | --- | --- | --- | --- | --- | --- | --- | --- |
| | | Men | Women | Men | Women | Men | Women | Men | Women |
| | | r | | r | | r | | r | |
| Men | High-intensity PA | 0.14 | 0.03 | 0.26 ** | 0.12 * | −0.20 ** | 0.02 | −0.04 | −0.07 |
| | Moderate-intensity PA | 0.04 | 0.05 | 0.05 | 0.01 | 0.17 * | 0.00 | −0.21 ** | −0.02 |
| | Low-intensity (walking) PA | 0.06 | 0.16 ** | 0.03 | 0.13 * | −0.08 | 0.14 * | 0.01 | 0.06 |
| | Total PA | 0.11 | 0.15 ** | 0.21 ** | 0.01 | −0.11 | 0.10 | −0.18 * | −0.02 |

Legend: ** Correlation is significant at the 0.01 level; * Correlation is significant at the 0.05 level.

The relationships between the level of physical activity and the domains of the quality of life of female teachers are shown in Table 3. High-intensity PA was significantly associated (although with a low correlation) with Psychological health (0.12; Sig. = 0.033), while Low-intensity (walking) PA was significantly associated with Physical health (0.16; Sig. = 0.004), Psychological health (0.13; Sig. = 0.027) and with Social relationships (0.14; Sig. = 0.012). The relationship between Total PA and Physical health was positive and relatively low (0.15; Sig. = 0.002).

### 3.1. Regression Analysis—Men

The analysis of the obtained results determined that there is a statistically significant influence of the predictor set (physical activity) on the criterion (quality of life) of male teachers in the domains of Psychological health (R = 0.35; $R^2$ = 0.12; F = 6.44, Sig. = 0.000), Social relationships (R = 0.32; $R^2$ = 0.10; F = 5.16, Sig. = 0.001), and Environment (R = 0.24; $R^2$ = 0.06; F = 2.64, Sig. = 0.021). The influence of physical activity on physical health is at the border of significance (R = 0.22; $R^2$ = 0.05; F = 2.34, Sig. = 0.056). The analysis of the influence of physical activity variables on the criterion was performed using the standardized regression coefficient beta (β) (Table 4).

**Table 4.** Partial impacts of the level of physical activity on the quality of life of male teachers.

| | Physical Health | | Psychological Health | | Social Relationships | | Environment | |
|---|---|---|---|---|---|---|---|---|
| | β | Sig. | β | Sig. | β | Sig. | β | Sig. |
| High-intensity PA | 3.73 | 0.025 * | 4.85 | 0.003 ** | 3.77 | 0.020 * | −0.13 | NS |
| Moderate-intensity PA | 3.27 | 0.022 * | 4.38 | 0.002 ** | 3.54 | 0.011 * | −0.33 | NS |
| Low-intensity (walking) PA | 2.87 | 0.020 * | 3.70 | 0.002 ** | 2.88 | 0.016 * | −0.18 | NS |
| Total PA | −4.63 | 0.021 * | −6.14 | 0.002 ** | −4.77 | 0.014 * | 0.08 | NS |

Legend: PA—physical activity; β—Standardized, partial, regression coefficient; NS—not significant; Sig.—the level of significance, ** $p < 0.01$, * $p < 0.05$.

The results of the standardized partial regression coefficient showed that all domains of PA affect Physical health ($p < 0.05$; High-intensity PA—β = 3.73; Moderate-intensity PA—β = 3.27; Low-intensity (walking) PA—β = 2.87; Total PA—β = 4.63), Psychological health ($p < 0.01$; High-intensity PA—β = 4.85; Moderate-intensity PA—β = 4.38; Low-intensity (walking) PA—β = 3.70; Total PA—β = 6.14) and Social relationships ($p < 0.05$; High-intensity PA—β = 3.77; Moderate-intensity PA—β = 3.54; Low-intensity (walking) PA—β = 2.88; Total PA—β = 4.77). However, it was discovered that there was no connection between any PA levels and Environment.

*3.2. Regression Analysis—Women*

On the sample of female teachers, the results showed that there is a statistically significant impact of physical activity on the quality of life in the domains of Physical health (R = 0.21; $R^2$ = 0.04; F = 3.43, Sig. = 0.009) and Psychological health (R = 0.18; $R^2$ = 0.33; F = 2.54, Sig. = 0.039). No statistically significant impact of the levels of PA on Social relationships (Sig. = 0.094) and Environment (Sig. = 0.198) was determined (results of partial impacts are not tabulated).

Standardized partial regression coefficients (Table 5) showed that there is no significant level of physical activity that affects Physical health ($p < 0.05$; High-intensity PA—β = 3.73; Moderate-intensity PA—β = 3.27; Low-intensity (walking) PA—β = 2.87; Total PA—β = 4.63) and Psychological health ($p < 0.01$; High-intensity PA—β = 4.85; Moderate-intensity PA—β = 4.38; Low-intensity (walking) PA—β = 3.70; Total PA—β = 6.14), that is, that only the entire system of predictors significantly affects the criterion.

**Table 5.** Partial impacts of the levels of physical activity on the quality of life of female teachers.

| | Physical Health | | Psychological Health | | Social Relationships | | Environment | |
|---|---|---|---|---|---|---|---|---|
| | β | Sig. | β | Sig. | β | Sig. | β | Sig. |
| High-intensity PA | −1.27 | NS | 1.09 | NS | −0.63 | NS | 1.51 | 0.048 * |
| Moderate-intensity PA | −1.59 | NS | 1.47 | NS | −0.83 | NS | 1.90 | 0.043 * |
| Low-intensity (walking) PA | −1.24 | NS | 1.38 | NS | −0.57 | NS | 1.71 | 0.036 * |
| Total PA | 2.461 | NS | −2.19 | NS | 1.26 | NS | −2.87 | 0.041 * |

PA—physical activity; β—standardized partial regression coefficient; NS—not significant; Sig.—the level of significance, * $p < 0.05$.

## 4. Discussion

The study's objectives were to determine whether PA levels have an impact on the quality of life of men and women working in the education sector and whether there are differences between the levels of Physical activity and quality of life domains among these professionals.

The findings show that, in comparison to males who engage more in Low-intensity PA, female teachers engage significantly more in High-intensity PA. The findings of this study are unexpected since women predominantly opt for low-intensity physical activity, such as walking, more frequently than other forms of activity (Kassavou et al. 2013; Đošić et al. 2021), whereas men prefer engaging in sports games and PA of high intensity (Hunt et al.

2014). More recent research conducted by Amagasa et al. (2022) showed that women are more physically active than men and spend less time engaging in sedentary activities. This could be clarified by the fact that increasing numbers of women engage in physical activity and that PA is becoming important to their way of life (Ou et al. 2017). Additionally, the study's male and female teachers' levels of physical activity were similar, which is different from other previous research findings that showed men reporting much higher levels of physical activity (Puciato et al. 2013; Al-Hazzaa 2018; Giustino et al. 2020).

The results also show that QoL is significantly higher in male teachers in the domains of Physical health and Psychological health than in female teachers. This is consistent with other studies where social norms and expectations frequently prioritize men's physical health and strength (Samulowitz et al. 2018). This can lead to increased opportunities for men of different professions to engage in physical activities, which positively impact their physical health (Janssen and LeBlanc 2010) and their perception of it (Dhuli et al. 2022). Male teachers tend to be more physically active than their female coworkers, and they seem to subjectively assess their psychological well-being to a greater degree, though this does not always imply accuracy. They may be discouraged by traditional gender roles from talking about their feelings or from receiving care for psychological problems, which could leave the impression that they report better psychological health (Lefkowich et al. 2017). While earlier research reported opposite findings (Sławińska et al. 2013; Louzado et al. 2021), the results of our study showed that female teachers outscored men in the Social relationships and Environment domains. This can be explained by the fact that female teachers may have larger social networks and increased frequency of social contacts due to their work activities in contrast to women from other professions. Women might also prioritize and put more effort into upholding and fostering social interactions, which can contribute to a greater sense of connectedness, support, and satisfaction in their social relationships, ultimately resulting in higher scores in the social relationship domain in our female sample of respondents. Additionally, women may have a heightened awareness of and sensitivity to their physical environment. They may place more importance on factors such as safety and security, physical settings (cleanliness, aesthetics) and availability of various resources, all related to the environment. Consequently, they may report higher scores in the environmental domain, which is also the case with our respondents, reflecting their satisfaction with their surroundings and the impact it has on their quality of life.

The study's findings demonstrate a positive association between PA and teachers' quality of life, similarly with the results of Marquez et al. (2020). Namely, there are significant connections between High-intensity PA and the QoL of teachers in the areas of Psychological health and Social relationships, as well as Moderate-intensity PA and Social relationships and Environment. The outcomes align with the conclusions of certain previous research studies (Krzepota et al. 2015; Puciato et al. 2018). The studies conducted by Krzepota et al. (2015) and Alexe et al. (2022b, 2022c) examined the relationship between High- and Moderate-intensity physical activity and Psychological health, finding that there are positive associations between PA and components like happiness and optimism in life, and negative associations between physical activity and anxiety and depression.

The obtained results indicate a connection between physical activity of moderate intensity and the social domain, which is in accordance with the study of Puciato et al. (2018). Fergus et al. (2019) showed that physically active men enjoy more fulfilling sexual lives, while the study by Marlier et al. (2015) pointed out that they also achieve interpersonal relationships more easily.

There is an association between Moderate-intensity PA and Environment among teachers. This quality-of-life area is the least studied, but there is evidence that Moderate-intensity PA with appropriate frequency is positively correlated with socioeconomic conditions (Puciato 2016), as well as with good environmental characteristics (Whitfield et al. 2018).

The established relationships of the PA levels with the domains of the QoL of female teachers indicate that High-intensity PA is significantly associated with Psychological health, while Low-intensity (walking) PA is significantly associated with Physical health,

Psychological health and Social relationships. The relationship between Total PA and Physical health is positive. A recent study (Patten et al. 2023), which determined the quality of life of women in relation to the intensity of physical activity, showed similar results. Women who trained with high-intensity physical activity had an increase in positive feelings and a decrease in negative feelings, anxiety, depression, and stress. In the aforementioned study, physical activity of moderate intensity contributed to the reduction in stress and negative feelings (Patten et al. 2023), and the study White et al. (2020) showed that the physical activity of walking applied at work is associated with better psychological health.

Moderate-intensity PA is also related to women's social interactions; that is, physically active women have stronger interpersonal relationships, more positive self-perceptions, and strong motivation to participate in activities (Coleman et al. 2008). Additionally, research indicated that women who engage in physical activity experience better sex lives (Dąbrowska et al. 2010).

Total PA and Low-Institutional Review Board Statement intensity PA are positively correlated with the Physical health domain. Female teachers often suffer from discomfort and back pain due to long periods of standing and sitting, and physical activity of moderate intensity can reduce discomfort (Durmus and Ilhanli 2012). Regular physical activity of moderate intensity was also associated with an increase in energy and fatigue in adult women (Zaslavsky et al. 2020).

Physical activity of both high and moderate intensity, as well as total PA, affect Physical health and Psychological health in teachers of both sexes. This is in line with the results of the study Knaeps et al. (2017) which concluded that regular physical activity of a larger volume, regardless of the intensity, affects the physical fitness associated with the health of adults. The adult population that is not physically active shows greater problems with sleep and rest than physically active individuals (Stea et al. 2022). The frequent application of physical activity by teachers affects the reduction in back pain and musculoskeletal pain in general (Moreira-Silva et al. 2016) as well as the reduction in sick days and the increase in work performance quality (Wynne-Jones et al. 2014).

Both male and female teachers' Psychological health is found to be positively impacted by PA. Physical activity increases positive thoughts and feelings, and decreases negative feelings, anxiety, depression, and stress in adult men and women (White et al. 2020; Alexe et al. 2022a; Patten et al. 2023). Adults who exercise regularly report better results in concentration and memory (Lambourne 2006), as well as self-esteem (Gilani and Dashipour 2017).

The impact of physical activity on the Social relationships domain is determined only among male teachers. When considering engagement in moderate or vigorous physical activity, it is crucial to consider potential variations in age, biological maturation, and the specificity of the physical activity performed, especially in terms of gender differences (Čaušević et al. 2023). Through physical activity, which often includes team sports and the participation of a large number of people, men make social contacts. The results are in line with previous studies, which point out that men who practice physical activity more easily make contact with others, possess improved public speaking abilities and have better sex lives (Marlier et al. 2015; Puciato et al. 2018; Fergus et al. 2019).

## 5. Conclusions

This study's results offer a new understanding of the relationship between PA levels and QoL in a specific group of professionals who are under the influence of a sedentary lifestyle resulting from their occupation. Based on the collected data, it is determined that whereas male teachers have a considerably greater level of physical activity of Low intensity, female teachers have a significantly higher level of High-intensity PA. Furthermore, although there is no discernible difference in the overall level of physical activity, the data indicate that male teachers are typically more physically active than female teachers. In the areas of Physical and Psychological health, male teachers have significantly higher quality of life, whereas women score higher in the areas of Social interactions and Environ-

ment. By understanding how different levels of physical activity affect their quality of life, teachers can make decisions about their own health and well-being, and potentially make adjustments to their lifestyle to optimize their overall quality of life.

The study has several limitations that should be considered. Firstly, the wide age range of participants may affect the interpretation of physical activity levels and quality of life, as different age groups may perceive these differently. Secondly, sociodemographic characteristics and cultural differences among participants may also impact the findings of the study. Additionally, the use of subjective methods such as questionnaires to assess physical activity levels and quality of life may introduce bias. Finally, factors such as comorbidities, medication use, and cultural differences could influence the results.

Further research is needed to explore these nuances and develop targeted interventions that promote physical activity as a means to enhance teachers' quality of life.

**Author Contributions:** Conceptualization, D.Ž. and S.P.; methodology, D.Ž.; software, S.P.; validation, A.Đ., T.P.-I. and M.Z.; formal analysis, L.M.; investigation, D.I.T.; resources, A.-M.V.; data curation, S.P.; writing—original draft preparation, D.Ž.; writing—review and editing, A.Đ.; visualization, C.S.; supervision, D.Ž.; project administration, D.I.T.; funding acquisition, C.I.A. All authors have read and agreed to the published version of the manuscript.

**Funding:** This research received no external funding.

**Institutional Review Board Statement:** The study was conducted in accordance with the Declaration of Helsinki, and approved by the Ministry of Education and Culture of the Republic of Srpska (No. 07.05/059-354-1/21) and the Pedagogical Institute (07/2.01/03614-103/21).

**Informed Consent Statement:** Informed consent was obtained from all subjects involved in the study.

**Data Availability Statement:** The dataset used and/or analyzed during the current study is available from the corresponding author in response to a reasonable request.

**Conflicts of Interest:** The authors declare no conflicts of interest.

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
