# Peer review of "Physical Activity and Quality of Life among High School Teachers: A Closer Look"

_socsci, doi:10.3390/socsci13030172_

Round 1
Reviewer 1 Report
Comments and Suggestions for Authors
Introduction
- Line 29-43 “Previous guidelines….” You should work on that sentence some. It is too full and different info. Becomes too difficult to follow.
- I agree that PA helps avoid health conditions, but you aren’t really dealing with that at all in this study so seems irrelevant to put a sentence about it here unless you’re going to talk about medical risk with teachers. Same with the statement about body image. I know these things are impacted by PA, but your study isn’t dealing with these variables.
Materials and methods
- This could be an editor issue, but why did you use commas in age, height, weight, and BMI? 37.4 should be a decimal place, not a comma 37,4. In Table 1, it should be body height too, not high.
- You should specify the timeline that the IPAQ assesses. Some readers may not know that it asks the previous 7 days of PA.
- I’d like a little more explanation of how you used MET levels here. You only talked about walking or generic MET levels. I know the IPAQ gives a rough estimate of METs. If you are going to use METs, you should really have the correct formulas to determine them for each person for each activity. Or if you’re using generic MET levels from IPAQ, you should state that and that they may not be as accurate as they don’t take into account patient’s body size.
Results
- End of first paragraph, you should include significance level for environment too since you did for the others.
-I think Table 2 could be much improved by putting some raw data numbers in there too, not just t values. Meaning put in the number of minutes of high intensity PA for men and women, score in physical health for men and women, etc. Then have the t score and significance levels next to those. It would all be one table but give the reader the ability to see the values and not just the statistical info.
- Table 3 paragraph needs the word “and” between high intensity PA and social relationships. The I in there reads strangely.
- I think it would be better to make Table 5 match Table 4 but obviously with the women’s data. Leaving part of the data out gives a sort of incomplete picture even though your writing does state all results for the women. Especially since you stated that women had higher values for relationships and environment.
Discussion
Last paragraph before conclusions – you state that only men had impact of PA on social relationships. Although they didn’t hit significance on regression, you stated there were correlations between PA and social relationships for women. Just make the discussions a bit more clear about these differences.
Comments on the Quality of English LanguageIt is good overall but there were some wording and grammar issues.
Author Response
To the reviewer 1. Thank you for your effort in evaluating our work and for the suggestions made to improve the article. You have attached the corrections made by us.

Reviewer 2 Report
Comments and Suggestions for Authors
Thank you for the opportunity to review this paper. I think that it is an interesting topic. Although I am unclear of the real-world impact that the results from this study would have this needs to be clarified. You note a difference between male and female teachers but you do not make it clear whether this means approaches to improve teacher physical activity and quality of life should be different between males and females. You also include no data on teachers who identify as neither male or female. The reason why this is the case needs stating in the manuscript.
Introduction
Line 29: The stated physical activity guidelines are out of date. These (https://cdnsciencepub.com/doi/full/10.1139/H11-009) are the most up to date Canadian One that I am aware of. Please note I am not based in Canada so their maybe newer ones. As this journal is for an internation audience it maybe better to quote the WHO guidelines instead.
Line 45: The fist sentence is a repeats what is discussed in the previous paragraph.
General comment: The introduction is a great summary of physical activity levels and physical activities relationship with quality of life. However it does not discuss in any detail why looking at the link between physical activity and quality of life in teachers is particularly important. Opposed to looking at the link in the working population more generally.
Methods
Line 94: Please indicate the age of pupils in a high school.
Table 1: Unsure what body high means I assume height
Move respondent demographics to results
Up to line 132: There is repetition in this section please remove.
Results
Table 2: I would remove the t values and replace with mean and confidence interval for both men and women.
Table 3: Can this be reorganised so that the men and women data for the same thing is on the same line so that it is easier to compare the numbers.
Regression analysis section: Please can the men and women sections be presented together.
Discussion
Second paragraph: Is a challenge to follow. Please consider rephrasing. I think what you are saying is that the findings from this study contradict some other studies but agree with some as well but this is unclear.
Third paragraph: Please relate the references back to the current study, currently it reads like a literature summary.
A more in depth discussion on the study weaknesses is required.
Author Response
To the attention of the reviewer 2.
Thank you for your effort in evaluating our work and for the suggestions made to improve the article. You have attached the corrections made by us.
